# GENERATIVE FAIRNESS TEACHING

## ABSTRACT

Increasing evidences has shown that data biases towards sensitive features such as gender or race are often inherited or even amplified by machine learning models. Recent advancements in fairness mitigate such biases by adjusting the predictions across sensitive groups during the training. Such a correction, however, can only take advantage of samples in a fixed dataset, which usually has limited amount of samples for the minority groups. We propose a generative fairness teaching framework that provides a model with not only real samples but also synthesized samples to compensate the data biases during training. We employ such a teaching strategy by implementing a Generative Fairness Teacher (GFT) that dynamically adjust the proportion of training data for a biased student model. Experimental results indicated that our teacher model is capable of guiding a wide range of biased models by improving the fairness and performance trade-offs significantly.

## 1 INTRODUCTION

Automated learning systems are ubiquitous across a wide variety of sectors. Such systems can be used in many sensitive environments to make important and even life-changing decisions. Traditionally, decisions are made primary by human and the basis are usually highly regulated. For example in the Equal Credit Opportunity ACts (ECOA), incorporating attributes such as race, color, or sex into credit lending decisions are illegal in United States (Mehrabi et al., 2019). As more and more of this process nowadays is implemented by automated learning systems instead, algorithmic fairness becomes a topic of paramount importance. Lending (Hardt et al., 2016), hiring (Alder & Gilbert, 2006), and educational rights (Kusner et al., 2017) are examples where gender or race biased decisions from automatic systems can have serious consequences. Even for more mechanical tasks such as image classification (Buolamwini & Gebru, 2018), image captioning (Hendricks et al., 2018), word embedding learning (Garg et al., 2018; Bolukbasi et al., 2016), and named co-reference resolution (Zhao et al., 2018), algorithmic discrimination can be a major concern. As the society relies more and more on such automated systems, algorithmic fairness becomes a pressing issue. Although much of the focus of developing automated learning systems has been on the performance, it is important to take fairness into consideration while designing and deploying the systems.

Unfortunately, state-of-the-art automated systems are usually data driven, which makes it more likely to inherit or even amplify the biases rooted in a dataset. This is an especially serious issue for deep learning and gradient based models, which can easily fit itself into the biased patterns of the dataset. For example, in a dataset with very few female candidates being labeled as hired in a job candidate prediction task, models might choose to give unfavorable predictions to qualified female candidates due to their under-representations in the training data. If deployed, such a biased predictor will deprive minority groups from acquiring the same opportunities as the others.

Much of the work in the domain of machine learning fairness has been focusing exclusively on leveraging knowledge from samples in a dataset. One straightforward way is to adjust the distributions of the training data through pre-processing. In the job candidate prediction example above, this means that we can either down-sample the majority class or up-sample the minority ones (Kamiran & Calders, 2012). Another family of fairness methods aims at matching the model performance on the majority class to that of the minority ones during training by using one of the fairness criteria (Gajane & Pechenizkiy, 2017). Some examples of such methods includes adding regularizations (Kamishima et al., 2012) or applying adversarial learning (Madras et al., 2018a). One issue with these approaches is that in many cases minority groups might be heavily under-represented in the dataset. Model training with fairness constraints will typically give up much of the performance ad-

vantages (e.g., prediction accuracies) in favor of the fairness metrics. Methods concentrate on solely on a dataset will often find themselves difficult to maintain a good performance - fairness trade-off.

One way to make models learn beyond the dataset is to take advantage of causal reasoning (Pearl et al., 2009), which borrows knowledge from external structures often formulated as a causal graph. Counterfactual Fairness (Kusner et al., 2017) and Causal Fairness (Kilbertus et al., 2017) are examples of such approaches. One unique characteristic of causal fairness is the fact that they need to be built based on a causal graph. And because those metrics are usually optimized and evaluated their own objective, which involves a causal graph, it's not clear how that added knowledge can be used to benefit other more commonly used fairness criteria such as Demographic Parity and Equalized Odds. Although it is possible to create causal structures that subsume conditional independencies in order to benefit DP or EO, we will need those structure information to be known in advance and we will have to derive one such structure for each metric we find. This is, what we believed, a significant limitation of the current causal methods which we aim to improve.

In this paper, we propose a generative approach for fairness training that is capable of leveraging both real data and "counterfactual data" generated from a causal graph. The counterfactual data is generated in a way that alters the sensitive attribute while keeping other latent factors unchanged. We formulate such generative model using a novel combination of adversarial training with mutual information regularization. Next, the two types of data are organized by an architecture called the teacher, which dynamically determines the proportion of real and counterfactual samples to train a particular model. Our model - Generative Fairness Teacher (GFT) can be used to improve an arbitrary fairness criteria based on need. Our experimental results indicate that we are able to take advantage of the counterfactual generative model and make it able to achieve a significantly better model fairness on a wide range of datasets across models. we are able to improve upon models with different levels of biases.

## 2 BACKGROUND

We provides a basic overview for the foundations of our method. Here we assume $X$ to be the input features, while $A$ being the set of sensitive features. We define $Y$ the be favorable outcome and $\hat{Y}$ to be the models' prediction of the favorable outcome given the features. The core idea of Fairness in machine learning is to distribute those favorable outcomes evenly across each of the sensitive group $A$.

### 2.1 FORMAL FAIRNESS CRITERIA

There has been many existing work on fairness focusing on studying criteria to achieve algorithmic fairness. A straightforward way to define fairness is Demographic Parity Madras et al. (2018a). In Demographic Parity, the chances of allocating the favorable outcomes $\hat{Y}$ is the same across sensitive groups $A$. Under that definition, the predictive variable $\hat{Y}$ is independent with $A$, making predictions free from discrimination against sensitive groups. Note that even though $A$ takes the form of a binary variable, we can easily extend the definition into the case of multiple values.

**Definition 1** *Demographic Parity*

$$P(\hat{Y}|X = x, A = a) = P(\hat{Y}|X = x, A = a')$$ (1)

Other fairness criteria that are built based on input features includes include Fairness Through Unawareness Gajane & Pechenizkiy (2017) , and Individual Fairness Kusner et al. (2017). More recently, Hardt et al. argued that criteria that only takes into account sample features making it difficult for the algorithms to allocate favorable outcomes to the actual qualified samples in both the minority and the majority groups. Such an observation leading to a new fairness criteria called Equalized Odds (and its special case Equal Opportunity) Hardt et al. (2016), where the fairness statement includes a condition on target variable $Y$.

**Definition 2** *Equalized Odds*

$$P(\hat{Y} = 1|X = x, A = a, Y = y) = P(\hat{Y} = 1|X = x, A = a', Y = y)$$ (2)

## 3 Causal Models and Counterfactual Examples

A causal model Pearl et al. (2000) is defined over a triple $(U, V, F)$ where $V$ is a set of observed variables and $U$ being a set of latent background variables. $F$ is defined to be a set of equations for each variable in $V$, $V_i = f_i(pa_i, U_{pai})$. Here $pa_i$ refers to the parent of $i$ in a causal graph. One importance concept in causal reasoning is intervention, in which case we substitute the variable of certain equation $v_i = v$.

We define a counterfactual example to be a synthesized sample generated from an existing data $X$ by manipulating its sensitive feature from $a$ to $a'$. Here we assume that both the real sample $X$ and the counterfactual sample $\hat{X}_{A \leftarrow a'}$ are generated from a latent code $U$.

**Definition 3** *Counterfactual Example*

$$\hat{X}_{A \leftarrow a'}(U)|X, a \tag{3}$$

### 3.1 Common Techniques for Fairness

Depending on when the fairness criteria are applied, methods for achieving fairness can be categorized as pre-processing, in-processing and post-processing Mehrabi et al. (2019).

**In-processing Techniques.** In-processing techniques apply fairness criteria during the training. Common techniques including fairness regularizer Kamishima et al. (2012) and adversarial training Madras et al. (2018a). Other methods fall into this category including the reduction based method Agarwal et al. (2018) and the more traditional discrimination approach in data miningHajian & Domingo-Ferrer (2012).

Our implementation applies in-processing techniques although our framework does not deal with in-processing methods directly.

**Pre-processing Techniques.** Pre-processing methods applies to the models before the actual training happens. Methods fall into this category are almost exclusively data processing techniques that aims at making the dataset free from biases. Re-sampling and re-weighting are two common techniques of pre-processing techniques for fairness. Calmon et al. (2017); Kamiran & Calders (2012); Agarwal et al. (2018). Other techniques include that repairs biases in a database Salimi et al. (2019). Our method is closely related to the pre-processing techniques because from the perspective of the student model our teacher model can be viewed as a data pre-processor.

**Post-processing Techniques.** When fairness adjustments are applied after the training is finished, techniques are called post-processing methods. Post-processing methods can be used to adapt models with all kinds of biases levels into a fair model Madras et al. (2018b). Other recently proposed including the method to model fairness as a score transformation problem Wei et al. (2019) and methods enforces Independence between sensitive features and model outcomes through Wasserstein-1 distances Jiang et al. (2020) Our approach is closely related to the post-processing technique as our teacher model can work with an arbitrarily biased student model.

## 4 Generative Fairness Teaching

In this section, we propose a teaching framework for training a student model that is able to work with a wide range of fairness criteria. We first present the overview of our approach in section 4.1. Then in section 4.2, we elaborate a novel generative model that can create "counterfactual examples". In section 4.4 we will show how to train such a teacher policy with the given student model and counterfactual generative model.

### 4.1 Fairness teaching framework

Given a training dataset $\mathcal{D}_{train} = \{(X = x_i, Y = y_i, A = a_i)\}_{i=1}^{|\mathcal{D}_{train}|}$, where $X$ and $Y$ are observed features and label, respectively, and $A$ is some sensitive attribute, we are interested in learning a predictive model $p_\theta(Y|X)$ that is parameterized by $\theta$, such that it maximizes the reward on the validation set:

$$R(\theta) = \mathbb{E}_{(x,y) \sim \mathcal{D}_{valid}} \log p_\theta(y|x) - \lambda_{fc}\text{FC}(p_\theta(Y|X), \mathcal{D}_{valid}) \tag{4}$$

---

**Algorithm 1** Generative Teaching Procedure

---

1: **Input:** initial student model $p_{\theta_0}(Y|X)$, counterfactual generator $\hat{p}(\hat{X})$
2: **Input:** dataset $\mathcal{D}$, Teacher policy $\pi_\psi$
3: **for** t $\leftarrow$ 1 to episode length $T$ **do**
4:     Sample a minibatch $D' = \{d_i = [x_i, y_i, a_i]\}_{i=1}^{M} \sim \mathcal{D}$.
5:     Get counterfactual data $\hat{D}' = \left\{\hat{d}_i = [\hat{x}_i \sim \hat{p}(\cdot|X = x_i, A = a_i), y_i, a_i']\right\}$ for each $d_i \in D'$.
6:     Get current student's state $s = S(D', p_{\theta_{t-1}}(Y|X))$ using Eq 11.
7:     Obtain decision $a_t \sim \pi_\psi(s)$, and get $\tilde{D} = \left\{d_i \in D'|a_t^{(i)} = 0\right\} \bigcup \left\{\hat{d}_i \in \hat{D}'|a_t^{(i)} = 1\right\}$.
8:     Update student: $\theta_t = \theta_{t-1} + \eta \nabla_{\theta=\theta_{t-1}} \frac{1}{|\tilde{D}|} \sum_{(x,y)\in\tilde{D}} \log p_\theta(y|x)$
9: **end for**
10: **Return:** updated student model $p_{\theta_T}(Y|X)$

---

where $FC(\cdot, \cdot)$ stands for the evaluation metric under certain fairness requirement, such as equalized odds. This objective tries to balance between the generalization error and the fairness constraint, which is controlled by the hyperparameter $\lambda_{fc}$.

In our teaching framework, we will teach a student predictive model $p_\theta(Y|X)$ to minimize Eq 4. Our teacher model $\pi_\psi$ is responsible for providing proper data samples for the student at each step of optimization to achieve this goal. In most teaching frameworks, the teacher is only responsible for selecting proper samples from existing dataset. However, due to the potential bias in the dataset, such assumption is too limited to achieve the fairness requirement.

Recall the definition of counterfactual example in Eq 3. Given a tuple of $(U, X = x, A = a)$, changing $A$ by $A \leftarrow a'$ while keeping $U$ fixed will also change $X$. Thus the change to the predictive distribution $p(\hat{Y}_{A\leftarrow a'}(U)|x, a)$ depends on two aspects: 1) the predictive model $p_\theta(Y|X)$, and 2) a *counterfactual generative model* $\hat{p}(\hat{X}) := p(\hat{X}_{A\leftarrow a'}(U)|X = x, A = a)$. Suppose we have the model $\hat{p}(\hat{X})$ ready, then it would be possible to regulate $p_\theta(Y|X)$ by generate counterfactual samples during training.

Given the teacher model $\pi_\psi$ and the counterfactual generative model $\hat{p}(\hat{X})$, we are ready to present our iterative teaching approach for learning $p_\theta(Y|X)$ in algorithm 1. At each teaching stage, the teacher will make binary decisions on using 1) the sample selection from the given dataset $\mathcal{D}_{train}$, or 2) the counterfactual data $(\hat{X}, Y, A = a')$ coming from data sample $(X, Y, A = a) \in \mathcal{D}_{train}$ and altered by $\hat{p}(\hat{X})$. The student will then use the selected samples to perform gradient update of $\theta$.

In the following sections, we will present how such counterfactual generative model $\hat{p}(\hat{X})$ are learned through teaching policy $\pi_\psi$.

## 4.2 LEARNING COUNTERFACTUAL GENERATOR $\hat{p}(\hat{X})$

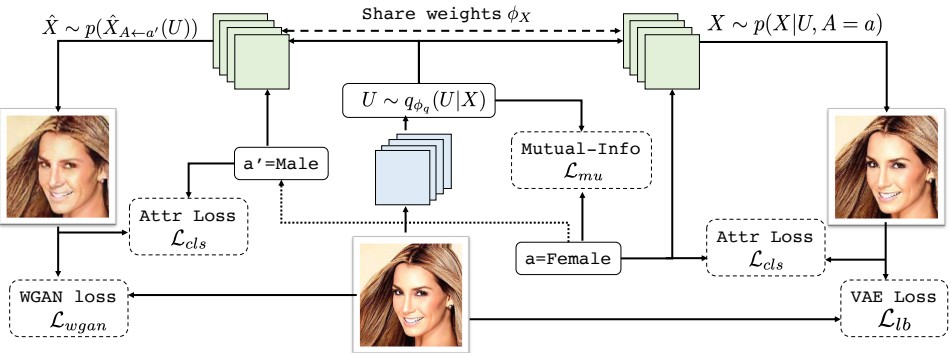

Figure 1: Overview of the counterfactual generative model.

To learn the counterfactual data distribution, we first need an understanding of the empirical data distribution. In next subsection, we first present our latent variable modeling of the data distribution:

### 4.2.1 EMPIRICAL DATA MODELING

We model the empirical data distribution as $p(X, A) = \int_U p(A)p(X|U, A)p(U)dU$. The design of this graphical model follows Kusner et al. (2017), where we have the dependency $U \to X \leftarrow A$. Here $U$ is assumed to be independent from sensitive attribute $A$, and $U$ and $A$ will become dependent when $X$ is observed. The generative process of counterfactual example $X$ depends on both $U$ and an altered $A$. To learn such latent variable model, we optimize the following lower bound:

$$
\begin{aligned}
\log p(X, A) \geq \mathcal{L}_{lb} \quad := \quad & \mathbb{E}_{q_{\phi_q}(U|X)} \left[ \log p(A)p_{\phi_X}(X|U, A) \right] \\
- \quad & D_{KL}(q_{\phi_q}(U|X)||p(U)), \text{ s.t. } I(A;U|X) = 0
\end{aligned} \tag{5}
$$

The mutual information constraint indicate that the posterior $q(U|X)$ should not be informative at predicting data distribution $p(A|X)$, and thus disentangles the sensitive and insensitive latent factors. We rewrite the mutual information in the following way:

$$
\mathcal{L}_{mu}(\phi_q) := I(A;U|X) = \mathbb{E} \left[ \log \frac{p(U, A|X)}{p(U|X)p(A|X)} \right] = \mathbb{E}_{q_{\phi_q}} \mathbb{E}_{p(A|X)} \left[ \log p(A|U;X) \right] + \mathrm{C} \tag{6}
$$

where the constant $C$ is the entropy $H(p(A|X))$ as $p(A|X)$ is the empirical data distribution. Thus suppose we have a predictive model $p_{\phi_A}(A|U)$ that is trained by minimizing:

$$
\mathcal{L}_{att}(\phi_A) = -\mathbb{E}_{X,A\sim\mathcal{D},U\sim q_{\phi_q}(U|X)} \left[ \log p_{\phi_A}(A|U) \right] \tag{7}
$$

then we can address the constraint in Eq 5 using penalty method, by minimizing $\mathcal{L}_{mu}$ w.r.t $\phi_q$.

### 4.2.2 COUNTERFACTUAL DATA GENERATIVE MODELING

As introduced in section 4.1, the counterfactual examples are generated by altering the sensitive attribute while keeping the latent factor $U$ unchanged. However we want to make sure these samples are also realistic, in the sense that it is close to the original data distribution $p(X)$. To match these two distributions, we leverage the technique of WGAN (Arjovsky et al., 2017) by optimizing the following objective:

$$
\mathcal{L}_{wgan}(\hat{p}(\hat{X}), D) = \min_{\hat{p}(\hat{X})} \max_{D} \mathbb{E}_{X\sim\mathcal{D}} \left[ D(X) \right] - \mathbb{E}_{(X,A=a)\sim D,U\sim q_{\phi_q}(U|X),\hat{X}\sim\hat{p}(\hat{X}_{A\leftarrow a'}(U))} \left[ D(\hat{X}) \right]
$$
$$ \tag{8} $$

where $D$ is the discriminator in GAN. We also adopt the gradient penalty (Gulrajani et al., 2017) $\mathcal{L}_{gp}$ with the discriminator to stabilize the training. Note that the counterfactual model shares the same decoder parameters as $p_{\phi_X}(X|U, A)$ in Eq 5.

We also leverage the attribute labels as auxiliary tasks for $D$. This auxiliary helps $D$ better distinguish between the realistic images and the generated counterfactual images. Here we create another linear layer on top of $D$'s last hidden layer (denoted as $D_A$) and try to minimize:

$$
\mathcal{L}_{cls} := \min_{D_A} -\mathbb{E}_{(X,A)\sim\mathcal{D}} \left[ \log p_{D_A}(A|X) \right] - \mathbb{E}_{\hat{X}\sim\hat{p}(\hat{X}_{A\leftarrow a'}(U)),a'} \left[ \log p_{D_A}(A = a'|X) \right] \tag{9}
$$

## 4.3 TRAINING GENERATIVE MODELS

As our generative models have multiple constraints with entangled dependencies (see Table 1 for summary), we design the following learning paradigm that can effectively satisfy these objectives:

a) Train the encoder $q_{\phi_q}(U|X)$, decoder $p(\phi_X)$, and discriminator $D, D_A$ in an alternating way, with $\mathcal{L}_a := \mathcal{L}_{lb} + \mathcal{L}_{wgan} + \mathcal{L}_{cls} + \mathcal{L}_{gp} + \mathcal{L}_{L2}(\phi_q, \phi_X, D, D_A)$ where the last term is the $L2$ regularization of neural network parameters;
b) Train the attribute classifier $p_{\phi_A}$ with $\mathcal{L}_{att} + \mathcal{L}_{L2}(\phi_A)$;
c) Finetune the encoder, decoder and discriminator with $\mathcal{L}_c := \mathcal{L}_a + \mathcal{L}_{mu}$.

Intuitively, the first step gets the generators working reasonably well in generating realistic images. The second step learns the attribute classifier from the latent code $U$, which is also a tractable task. In

Table 1: Notation of loss terms.

| Notation | Objective |
|----------|-----------|
| $\mathcal{L}_{lb}$ | generative modeling of $(X, A)$ with latent $U$ |
| $\mathcal{L}_{mu}$ | separating the information of $A$ from $U$ |
| $\mathcal{L}_{att}$ | learning the attribute classifier from $U$ |
| $\mathcal{L}_{wgan}$ | learning the counterfactual generator |
| $\mathcal{L}_{cls}$ | auxiliary task for attribute prediction |
| $\mathcal{L}_{gp}, \mathcal{L}_{L2}$ | gradient penalty and L2 regularization |

the last step, we address the mutual information constraint using penalty method, by minimizing the Eq 6 together with other generative model losses. The counterfactual generator would be expected to learn to adapt to the refined U in the last step. See Figure 1 for the visual demonstration of this process. In practice, we can also tune the coefficients of each loss term. See the experiment section for more information.

## 4.4 LEARNING TEACHER POLICY $\pi_\psi$

Since the student objective defined in Eq 4 is complicated, which involves with arbitrary fairness metrics, we leverage Policy Gradient to learn the teacher model $\pi_\psi$. Specifically, our objective is

$$\max_{\pi_\psi} \mathbb{E}_{\tau \sim \pi_\psi} \left[ \sum_{(s_t, a_t) \in \tau} r(s_t, a_t) \right] \tag{10}$$

where $\tau$ is the state-action trajectory sampled from the behavior $\pi_\psi$. Next we define the state, action and reward in detail. The RL-based teacher act as a data loader in the iterative learning process between teacher and student. It will feed the real or fake images to student according to current state. Following the teacher's instruction, the student model will have a final terminal reward on the held-out validation set.

- **state:** It contains the information of student's model and current training batch. We denote it as

$$S(D', p_\theta(Y|X)) = [\underbrace{\{y_i \in D'\}}_{\text{labels}}, \underbrace{\{a_i \in D'\}}_{\text{sensitive attributes}}, \underbrace{-\{p_\theta(y_i|x_i)\}}_{\text{cross entropy}}, \underbrace{\text{FC}(p_\theta(Y|X), D')}_{\text{group fairness on current training batch}} ] \tag{11}$$

- **reward:** The reward function defined in Eq 4, which is evaluated on held-out validation set. For example, if the Equalized Odds is used as fairness metric, then $FC(\cdot, \cdot) = -log(|P(\hat{Y} = 1|A = 1, Y = y) - P(\hat{Y} = 1|A = 0, Y = y)|)$. One can also define the reward for Demographic Parity or other fairness criteria.
- **action:** The teacher needs to make binary decision $\{a_m\}_{m=1}^M, a_m \in \{0, 1\}^M$ on the minibatch of instances, where $M$ is the batch size. Here 1 represent using real data, 0 represent using the corresponding counterfactual data sample generated from $\hat{p}(\hat{X})$.

The episode length typically equals to several epochs of the training data. We use the moving average of the final reward as the baseline to reduce the variance. Note that other advanced RL algorithms or techniques that handle delayed reward (Arjona-Medina et al., 2019) can also be adapted here to further boost the performance.

## 5 EXPERIMENTS

### 5.1 TABULAR DATA

**Experiment Details.** We perform binary classification and fairness metric analysis on tabular data to show the improvement of performance using Generative Fairness Teacher (GFT). Prediction performance is measured by Testing Error. We choose Equalized Odds defined in Eq.2 to be our fairness metrics to illustrate the performance of our model. In practice one can choose to optimize an arbitrary metrics based on needs. In each of the experiments we evaluate the gap of Equalized Odds, defined as

$$EO = P(\hat{Y} = 1|X = x, A = a, Y = y) - P(\hat{Y} = 1|X = x, A = a', Y = y) \tag{12}$$

| Table 2: Adult Dataset | | |
|---|---|---|
| Method | Error(%) | EO |
| In-processing | 16.8 | 0.048 |
| Post-processing | 16.9 | 0.049 |
| Base1: all real | 15.9 | 0.179 |
| Base2: all fake | 15.8 | 0.141 |
| Base3: random | 16.0 | 0.180 |
| Base4: balance | 15.8 | 0.157 |
| GFT | 16.1 | **0.044** |

| Table 3: COMPAS Dataset | | |
|---|---|---|
| Method | Error(%) | EO |
| In-processing | 32.9 | 0.034 |
| Post-processing | 32.8 | 0.039 |
| Base1: all real | 32.1 | 0.224 |
| Base2: all fake | 32.3 | 0.231 |
| Base3: random | 32.7 | 0.247 |
| Base4: balance | 32.0 | 0.229 |
| GFT | 32.8 | **0.028** |

| Table 4: CelebA Dataset | | |
|---|---|---|
| Method | Error(%) | EO |
| Base1: all real | 21.0 | 0.426 |
| Base2: all fake | 48.1 | 0.610 |
| Base3: random | 23.5 | 0.417 |
| Base4: balance | 19.6 | 0.362 |
| Base5: fix ratio | 20.7 | 0.242 |
| Base6: reverse | 19.2 | 0.171 |
| GFT | **19.1** | **0.098** |

We report the maximum among the false positive difference and true positive difference between protected and unprotected groups. We compared our method with exponentiated-gradient reduction based in-processing algorithm (Agarwal et al., 2018) and score-based post-processing algorithm (Hardt et al., 2016). In addition to these two methods, we also compared the GFT with four different baselines. Base1 denotes the model that trains with all original examples, which is also an unconstrained classifiers. Base2 denotes the model that trains using all counterfactual examples. Base3 is the model that trains with a random combination of original and counterfactual examples. Base4 is the model trained with a balance combination of original and counterfactual examples, which guarantees the proportion of protected and unprotected group in the training set to be the same. We evaluate our method on two well-known tabular datasets, the ProPublica's COMPAS recidivism dataset and the UCI Adult income dataset. The model are trained on randomly selected 75% samples and evaluated on the rest of 25% testing examples. We follow the same setting as in (Agarwal et al., 2018), which uses logistic regression in scikit-learn as the classifier.

**Adult Income Dataset.** The Adult Income dataset contains information about individuals from the 1994 U.S. census. There are 48,842 instances and 14 attributes, including sensitive attributes race and sex. From Adult dataset, we select age, education number of years, relationship, race, sex, capital-gain and hours-per-week to be the decision variables. The binary classification task here is to predict whether an individual makes more or less than $50k per year. The results in Table 2 show that our GFT can achieve the lowest EO score with the minimum sacrifice on the testing error compared with other fairness algorithms, achieving the best performance - fairness trade-offs. Additionally, GFT outperforms the four baselines, indicating that our generative fairness teaching is indeed more effective than combing the original and the counterfactual data in a mechanical way.

**COMPAS Dataset.** The ProPublica COMPAS dataset has a total of 7,918 instances, each with 53 features. From COMPAS dataset, we select age, race, sex, count of prior offences, charge for which the person was arrested and COMPAS risk score to be the decision variables. The binary target outcome is defined as whether or not the defendant recidivated within two years. The ProPublica COMPAS dataset has a total of 7,918 data instances, each with 53 features. From COMPAS dataset, we select age, race, sex, count of prior offences, charge for which the person was arrested and COMPAS risk score to be the decision variables. Experimental results are illustrated in Table 3, where one can observe the GFT is consistently better than other methods in terms of Equalized Odds. Similar to the results in Adult income, we see that GFT achieved the best performance - fairness trade-offs among all of the methods tested.

## 5.2 Image data

**Experiment Details.** We also evaluate our GFT on visual recognition task. In order to generate high quality counterfactual examples, we leverage an U-Net like connections between encoder and decoder. The adversarial classifier implemented on the latent representation of the image is a stack of 9 convolution layers followed by fully connected layers. We follow the same baseline settings as in section 5.1 to combine original and counterfactual images, the other two fairness algorithms are not applicable here due to the format of image data. In addition to the four baselines, we add two different settings. Base5 denotes the baseline model that trains on 90% counterfactual examples and 10% original examples. Training data in Base6 is obtained by maintaining a part of the original examples and adding counterfactual examples to reverse the original biased distribution among protected and unprotected group. The EO score we use here is the sum of the false positive difference

and true positive difference between protected and unprotected groups. In the visual recognition task, the student model is a VGG-16 network trained using momentum SGD optimizer.

**CelebA Dataset.** CelebA is a commonly used large-scale face attribute dataset. There are 202,599 images, each with 40 binary attributes that reflect appearance (hair color and style, face shape, makeup, for example), emotional state (smiling), gender, attractiveness and age. For this dataset, we use 'Gender' as the binary sensitive attribute. Among the other 39 attributes, We choose one of the most correlated attributes to 'Gender', the 'Arched_Eyebrows' as our classification target to make this task more challenging. As shown in Table 4, GFT reduced the EO score significantly comparing to the baselines. We also achieved $\sim 0.9\%$ improvements on testing error, outperforming the balanced baseline.

**Improving Fairness across Models with Different Bias Level.** We perform a post-processing teaching experiment on four different biased student models. A is the sensitive attribute and Y is the classification target. We manually select four pre-training image sets according to the certain ratio shown in Figure 2. After training on these dataset respectively, we obtained four student base models with different unfair level. The gap of Equalized odds trend in Figure3 shows that the GFT is capable to alleviate the unfairness of various student base models in a post-processing manner.

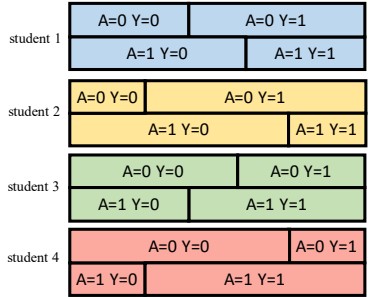

Figure 2: Different Unfair Ratio

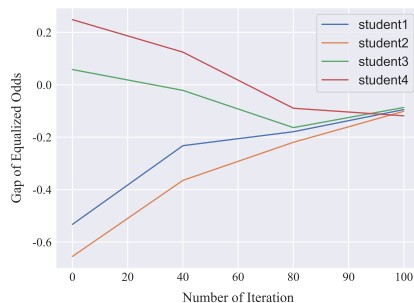

Figure 3: Post-Processing Result

## 5.3 ANALYSIS

**Counterfactual Examples.** We include a qualitative evaluation of our counterfactual generator in Fig.4. These visualizations demonstrate the difference of original images and the counterfactual images by manipulating the binary attributes. We choose the male, young and blonde hair as the sensitive attributes to show the effects of manipulating a specific property. One can observe that the target attribute 'Arched_Eyebrows' in our recognition task is not visually altered between the original example and the counterfactual one. Powered by a generative backbone, our counterfactual examples are of high quality.

**Teacher Model.** We analyze the training dynamics and the teaching behavior of our GFT model in this subsection for the CelebA dataset. We implement a policy gradient based teacher agent as the data loader to student agent. We shoe negative log reward in Figure 5, the training reward here is the final Equalized Odds score on the CelebA held-out validation set. After 50 episodes, the corresponding EO score will be smaller than 0.20, the unfairness is alleviated compared with the unconstrained baseline 0.67. Figure 6 demonstrates the action adopted by the teacher model. We also demonstrated the percentage (moving average with sample size 7) of original image in training the student model. As we defined in section 4.4, the teacher will make binary decision of whether to feed original or counterfactual image to student on each iteration. Since the teacher model interacts with student model during the training process, we observe that the percentage of using original image is also changing. As training progresses, there has been a gradual decline in the use of original images (and thus an increase in the counterfactual ones).

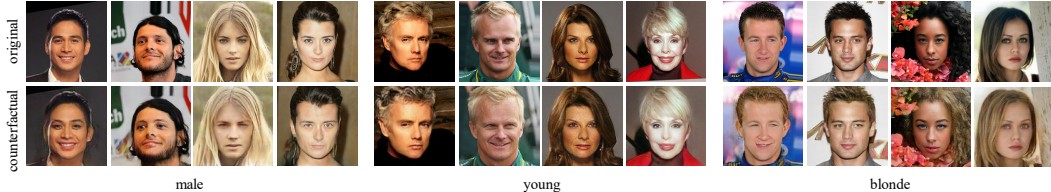

Figure 4: Examples of the counterfactual images on CelebA from the male, young and blonde_hair attribute. These result are obtained by our counterfactual generator.

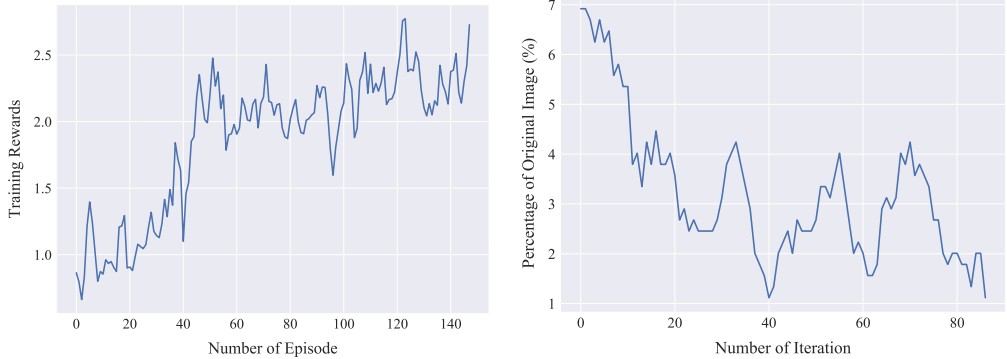

Figure 5: Training dynamic

Figure 6: Teacher action

## 6 CONCLUSIONS

In this paper, we propose the Generative Fairness Teaching (GFT) framework to achieve algorithmic fairness for machine learning models. Our method can generate high quality counterfactual examples, which is a novel approach to compensate the biases in a dataset. Together with a student - teacher architecture, we dynamically adjust the proportion of counterfactual examples and mix it with the original ones in order to train a fair model. Experimental results indicated that our method strongly out-perform baseline methods in both tabular and real image datasets.

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

# Appendix

## A    EXPERIMENTAL DETAILS

| Encoder | Decoder | Discriminator | Adv. Classifier |
|---|---|---|---|
| Conv1 [4×4, 64, 2] | DeConv1 [4×4, 1024, 2] | Conv1 [4×4, 64, 2] | Conv1.x [3×3, 512, 1]×3 |
| Conv2 [4×4, 128, 2] | DeConv1 4×4, 512, 2] | Conv2 [4×4, 128, 2] | Conv2.x [3×3, 256, 1]×3 |
| Conv3 [4×4, 256, 2] | DeConv1 [4×4, 256, 2] | Conv3 [4×4, 256, 2] | Conv3 [4×4, 256, 1] |
| Conv4 [4×4, 512, 2] | DeConv1 [4×4, 128, 2] | Conv4 [4×4, 512, 2] | Conv4 [1×1, 128, 1] |
| Conv5 [4×4, 1024, 2] | DeConv1 [4×4, 3, 2] | Conv5 [4×4, 1024, 2] | Conv5 [1×1, n_class, 1] |

Table 5: Our counterfactual generative model architecture. Conv1.x, Conv2.x and Conv3.x denote convolution units that may contain multiple convolution layers. E.g., [4×4, 64, 2]×3 denotes 3 cascaded convolution layers with 64 filters of size 4×4 and stride 2.

**Generative model settings.** The network architectures used in the paper are elaborated in Table 5. There is an U-Net like connections between Encoder Conv4 layer and Decoder DeConv1 layer. Except the Adversarial Classifier, which uses e Stochastic Gradient Descent Optimizer with momentum 0.9. The other module use the Adam Optimizer. We use batch size 64 and start with learning rate $1e-4$. We will first pre-train the Generator and Discriminator for 60 epochs and fix, then train the Adversarial Classifier solely for 20 epochs, finally we fine-tune Generator and Discriminator again for another 60 epochs.

**Teacher and student model settings.** The network architecture used in the teacher model is a three-layer neural network with layer size $d \times 15 \times n$ , where $d$ and $n$ represent the dimension of state and number of action. VGG-16 is used in the face attribute classification task as the student model. We train the teacher model for 500 episodes, within each episode, the student model is re-initialized and trained for 20 epochs. Teacher model and student model are optimized by Adam and Stochastic Gradient Descent Optimizer with momentum 0.9 respectively. We start with learning rate $1e-3$ for Adam and 0.1 for momentum SGD, divide it by 10 when the performance is saturated. We use batch size 64. The terminal reward is measured on the held-out validation set after 20 epochs of student training, the final result is measured on the testing set.

**Tabular dataset settings.** For the tabular data experiments, we follow the same settings in the original paper (Agarwal et al., 2018) and the official repositories.[1] These settings also include the standard data pre-processing steps, which convert the data into suitable format for the ML algorithms. Then the data is randomly split into training and testing set in a ratio of 75% and 25%. The only different setting is the choices of the decision variables. We use age, education number of years, relationship, race, sex, capital-gain and hours-per-week to be the decision variables in the Adult dataset. We use age, race, sex, count of prior offences, charge for which the person was arrested and COMPAS risk score to be the decision variables in the COMPAS dataset.

---

[1]https://github.com/fairlearn/fairlearn

