# OpenReview forum: "Generative Fairness Teaching"
_ICLR.cc/2021/Conference — Reject_

### Official Review · AnonReviewer2 · 2020-10-24
**An interesting work**

**Rating:** 6
**Confidence:** 2

**Review:**

This paper describes a pre-processing method to reduce certain statistical disparities in the classifier obtained from the training data. The proposed approach involves learning a latent probability model that simulates the training data. The authors then manipulate the learned model to generate "counterfactual" samples that belong to the membership of underrepresented demography. A more "fair" classifier is trained on the manipulated data mixing with the "counterfactual" samples.

The proposed approach seems sensible. The experiment results support the claims that it reduces the statistical disparities (e.g., equalized odds) of the trained classifier. I like the simplicity of the proposed method while future work is needed to explicate the theoretical condition under which it is sufficient.

Also, I believe that the clarity of this paper could be improved. For instance, the definition of counterfactual fairness seems to be orthogonal to the "counterfactual" samples described in later sections. More specifically, the counterfactual fairness concerns with the potential outcome of prediction $\hat Y_{a'}$ has the sensitive attribute $A$ been $a'$. On the other hand, the "counterfactual" samples is the potential outcomes of the feature $X_{a'}$ had $A = a'$. It does not seem that the concept of counterfactual fairness has be used in later sections. Therefore, I would suggest that the authors could remove the definition of counterfactual fairness and provide a formal introduction to structural causal models and interventions, e.g., see (Causality, Pearl, 2009).


------------ Post Rebuttal ------------------
I read other reviewers' comments and the authors' responses. I like the idea of applying the GAN framework to compute counterfactual distributions. However, I could also see why other reviewers are not particularly excited about it. The authors managed to apply the GAN approach to obtain counterfactual samples in some specific datasets. However, many questions regarding the proposed methods are left unanswered, e.g., under which condition the proposed GAN approach is ensured to obtain unbiased counterfactual samples. With this being said, I think this paper could be most improved by further elaborating how it contributes to the existing causal inference literature, especially in computing counterfactual probabilities. Due to these reasons, I intend to keep my score but won't strongly champion for it.

---

> ### Author Response · Authors · 2020-11-25
> **Reply to Reviewer 2**
>
> We thank the reviewer for the constructive comments. We do believe that the proposed approach can overcome some of the drawbacks of the existing counterfactual fairness training and have much room for future work.
>
> We also thank the reviewer for the feedback on the quality of the paper. Since the release of the review we have made a comprehensive review of the technical part of the paper and have improved many of the parts that may cause confusion, including:
> - Added graphical model we used in Sec 3.2.1
> - More explanations on our learning paradigm, including the auxiliary loss, the summarization of loss terms in Table 1.
> - A detailed description of RL based teaching method in Sec 3.4
> - More experimental details in Sec 4.2 and Appendix A.
>
> Please refer to the revised manuscript to see our changes.
>
> We have adopted the reviewer’s suggestion by removing the definition of counterfactual fairness in formal fairness criteria. Additionally, we’ve added another section called ‘Causal Models and Counterfactual Examples’ in the revised paper. In this section, we presented a formal definition of causal models along with the definition of counterfactual example. We have also removed the majority of the occurrence mentioning counterfactual fairness throughout the paper with the single exception of the introduction.

---

### Official Review · AnonReviewer4 · 2020-10-28
**Interesting direction, but more needs to  be done**

**Rating:** 5
**Confidence:** 4

**Review:**

The paper proposes a teacher-student framework to ensure fairness by letting the teacher choose examples for the student from either the training data or from a counterfactual distribution. The main contributions are a counterfactual generative model and an algorithm for learning the teacher policy.

Strengths:
========
1. The idea seems interesting and the proposed teacher-student framework is novel in the area of fair learning.

2. The authors have done a good job of modeling various aspects of their complex approach using neural networks.

3. The fact that authors were able to make the complex optimization work is itself a good thing since the objective has a lot of moving parts.

4. The presented evaluations also show some promise.

Things to consider improving:
=========================
1. My basic question is regarding the motivation for such a framework. Why is this approach important? Is there any fundamentally new insight that can be obtained with a GFT framework that cannot be obtained using existing fair learning approaches.

2. The argument that causal methods cannot benefit commonly used fairness metrics like demographic parity (DP) or equalized odds (EO) should be elaborated. Isn't it possible to create Structural causal models that subsume conditional independences like DP and EO?

3. There are a number of issues with references:
- There are a number of additional pre-, in-, and post-processing methods that can be cited. Looking into some recent fairness papers may give the authors an idea of the works.
- Agarwal et al. 2018 is considered an in-processing approach and not post-processing or pre-processing since it imposes fairness criteria while training a fair model.
- Demographic parity is a really old concept and has been discussed in many papers before Madras et al. 2018.
- The authors should also consider citing and perhaps comparing with https://homes.cs.washington.edu/~suciu/sigmod-2019-fairness.pdf if it makes sense, since it also does "database repair" or pre-processing.
- Barring some exceptions, many early works in fairness are not cited. For example https://ieeexplore.ieee.org/document/6175897 is a classic work that discusses a lot of concepts in fairness that we use today (they call it discrimination). I recommend the authors to do a thorough literature survey and include at least the important works.

5. Many crucial algorithmic details are missing. While  the complex optimization is important to cover all the aspects of the framework for data generation, it should also be motivated and explained better.
- In Sec. 3.2.1, how valid is it to assume U to be independent of A? Technically A is a part of the data that is generated so it may be reasonable to surmise that it should depend on the latent. In general the authors should provide a DAG which encodes their assumptions and justify them.
- How would this architecture be modified for DP since the authors discuss both DP and EO in the beginning?
- What is L_att trying to optimize?
- What does it mean to have attribute labels as auxiliary tasks for D (in L_cls)? What are attribute labels?
- It may make sense to summarize in a few sentences what each term in the objective does. If space  is less, the authors can move some contents to the appendix.
- In Sec. 3.3, could the authors provide some insights on why such a training works?
- Please also discuss what REINCFORCE is, and provide reference/more details.

6. Generally in experiments, cross-validated results are needed. It is also crucial to provide sufficient pre-processing and hyper-parameter details to help reproduction of results.

7. There are also a number of recent post-processing methods that can be compared with besides Hardt et al, 2016 (see http://auai.org/uai2019/proceedings/papers/315.pdf, http://proceedings.mlr.press/v108/wei20a.html)

8. More details needed on how Agarwal et al., 2018 was run for example. In https://arxiv.org/pdf/1906.00066.pdf, pg. 30  Agarwal seems to be more competitive than what is shown here. Look at adult-gender-LR-EO and  adult-gender-GBM-EO, Agarwal et al. 2018 (named "red"")  at EO=0.04 has higher  accuracies than shown here. Similar comments apply to COMPAS.

9.There are  also a few other pre-processing approaches that use GANs.
https://arxiv.org/pdf/1805.11202.pdf
https://arxiv.org/pdf/1805.09910.pdf

10. The authors must identify the caveats  of training a model on CelebA which has a "western" and "celebrity" bias. Models trained there may not transfer to other general settings.

11. In page 8, the analysis of teacher model needs more details. How can we say original images dominate if only 7% image is chosen at max at any iteration? Why is the teacher behavior of choosing real samples in the beginning and synthesized samples later justified?

In summary, the paper has identified an interesting direction but this needs to be taken forward a bit more.

Post-rebuttal
===========
Thanks to the  authors for  their detailed response to  my  questions. Some of the answers are indeed satisfactory, but some questions remain - such as extensive comparisons to other methods (probably using more datasets), how  the method would  behave (practically)  with  a different fairness measure like DP, and more  carefully situating the method in  the  fairness literature. I encourage the authors  to keep pursuing this interesting  direction.

---

> ### Author Response · Authors · 2020-11-25
> **Reply to Reviewer 4, part 1/3**
>
> Thanks a lot for the constructive feedback! We have addressed them correspondingly in the paper, please also see our response below:
>
> ### Q1: New insights and significance of the work
> ---
> The significance of the work lies in the fact that we demonstrated that fairness training can benefit from counterfactual examples generated by a learned generator. We also show that the training can benefit from a carefully tuned ratio of counterfactual examples and real examples as training progresses. Those are the insights we contributed to and are something that related work such as counterfactual fairness can not achieve, as they were using a fixed generator and a fixed counterfactual-real ratio.
>
> The optimization process of our generator is built on a student-teacher framework. Although much of the generative models are inspired by GAN, as we have illustrated in the response 9 of this rebuttal to the reviewer, our approaches and problem settings are fundamentally different from work that aims at improving the fairness of GAN models such as FairGAN. For those settings, GAN is chosen to be a problem while our generative models are used as a solution. Our approach is generanic to any fairness problem and can be applied to improve the fairness of a wide range of problems that is not limited to GAN models. To the best of our knowledge, we are the first work in the area to demonstrate that a combination of 1) a learned generator for synthesizing counterfactual examples and 2) a dynamic counterfactual-real ratio would benefit fairness training.
>
>
> ### Q2: Applicability of causal methods to more common fairness metrics
> ---
> We agree with the reviewer that it is possible to create causal structures that subsume conditional independencies in order to benefit DP or EO. However, we believe the arguments in the paper are still valid, as we will need those structure information to be known in advance. If we want to make those causal structures specific to the metrics, we will have to derive one structure for each metric we find. This is, what we believed, a significant limitation of the current causal methods which we aim to improve. To acknowledge the point that the reviewer is mentioned, we have updated the paper to clarify our arguments.
>
> ### Q3: Issues with references
> ---
> - Additional citations for additional pre-, in-, and post-processing methods
>
> We have moved Agarwal et al. 2018 to in-processing as the reviewer suggested.
> We have also incorporated the database repair paper along with the discrimiantion paper into the related work.
>
>
> ### Q5. “Many crucial algorithmic details are missing…”
> ---
> Please see our explanation below. We’ve also made refinements in the paper.
>
> *Q5.1: In Sec. 3.2.1, … assume U to be independent of A*
>
> We follow the graphical model from Kusner et.al (see Fig 1(a)(b)  in https://papers.nips.cc/paper/2017/file/a486cd07e4ac3d270571622f4f316ec5-Paper.pdf). It is standard to assume A -> X <- U. This is known as “v-structure” in directed graphical model, where observing X will couple A and U. For example, gender should be independent of educational background, but observing a picture X that shows a girl wearing a doctoral hat would couple the “female doctor”.
>
> *Q5.2: how ...modified for DP*
>
> We only need to modify the reward function. In the paper we use EO, but it is straightforward to replace it with DP objective, as the reinforcement learned teacher can handle the black-box reward function.
>
> *Q5.3: what is L_att*
>
> L_att learns an attribute predictor from U. The learned predictor will be used in Eq (6), where the latent factor U will try to minimize the mutual information between A, so as to decouple the sensitive information A for predicting target label Y.
>
> *Q5.4: What … for D (in L_cls)? What are attribute labels?*
>
> Each training sample is a  <X, A, Y> triplet, for example, X is an image, A is the gender, and Y is the hair style. The auxiliary task of predicting attributes using D’ on top of D is to help D better distinguish between realistic v.s. fake images.
>
> *Q5.5: summarize … each term in the objective*
>
> We’ve added a table that summarizes the roles of different losses in the objective.
>
> *Q5.6: In Sec. 3.3, ... why such a training works?*
>
> Firstly it is expected that learning GAN to generate realistic images would work in general. Also learning an attribute classifier would also be a tractable task. So our training is designed to first train these two parts (Sec 3.3 (a)(b)) separately, to provide a warm start.
> The procedure in (c) further minimizes the mutual information between U and A given X, and the counterfactual generator would also learn to adapt to the refined U. Since the generator is pretrained, the adaptive refinement procedure would be smoother.
>
> *Q5.7: discuss REINFORCE*
> The policy gradient is a standard RL approach. Experimentally we found the variance reduction is not very helpful, so we didn’t use actor-critic. We’ve added the explanation in the paper.
>
> **1/3 OF REPLY**

---

> ### Author Response · Authors · 2020-11-25
> **Reply to Reviewer 4, part 2/3**
>
> ### Q6: Pre-processing and hyper parameter details
> ---
> Thank you for pointing this out, please refer to the revised Appendix for more details.
>
> ### Q7: “There are also a number of recent post-processing methods..”
> ---
> Thanks for pointing out these papers. The two papers are indeed relevant and we have included in our paper for discussion. Although the two referenced papers used wasserstein distance, there are major differences compared to our paper: Jiang et.al learns to transport predictive distribution p_a to p* for each a, while Wei et.al learns to transform the score r’(x), which is in the reweighting regime. What we are doing is to transform p(X) to p_{A<-a’}(X), e.g.., the generative model of images. This way we can go beyond the images from the original dataset. Technically this is also significantly harder than Jiang et.al or Wei et.al. It is intractable to get the closed form transport, due to the high dimensional space (e.g., 1024-dimensional for 32x32 images).
>
> ### Q8. Difference in evaluation results for baselines
> ---
> We follow the implementation of Agarwal et al., 2018 in their original paper (https://arxiv.org/pdf/1803.02453.pdf). The only different setting is the choices of the decision variables. As introduced in section 4.1, for in-processing, post-processing and our methods, we only use age, education number of years, relationship, race, sex, capital-gain and hours-per-week to be the 7 decision variables in the Adult dataset, while 14 variables are used in https://arxiv.org/pdf/1906.00066.pdf. Similarly, we only use age, race, sex, count of prior offences, charge for which the person was arrested and COMPAS risk score to be the decision variables in the COMPAS dataset. It should be the reason that caused the accuracy gaps. Since our generative model is more suitable for image data and tabular data with fewer rows, consistent and fair comparison was the main reason we use the same decision variables in the in-processing and post-processing method. While our work focuses on the entire framework, there are techniques that could be applied to improve the tabular data generation, we will add experiments by modifying our generative model to better accommodate tabular data.
>
> ### Q9. Differences between our work and few other pre-processing approaches that use GANs
> ---
> There is a fundamental difference between our work and the line of work that improves fairness in the GAN setting. The lines of work such as FairGAN aim at improving fairness on GAN models. In other words, GAN is used as a problem setting rather than a solution. The purpose of our work, however, is to improve fairness in the general sense with the help of a technique that involves generative models and the idea of counterfactual reasoning. In this case, we are not really restricting the problem space to be GAN, but rather use the core idea of generative models as a solution. To avoid further confusions, we have clearly illustrated this difference in an updated version of the paper.
>
> ### Q10. Caveats of ‘western’ and ‘celebrity’ bias
> ---Our method is a general framework, it is not particularly designed for specific dataset. We could extend it to fit into other general settings. Such extensions include adding the number of teacher actions(flip both the gender and the race) or enriching the training and validation data with our generative model. In practice, tasks related to face attribute classification could be trained and validated on a diverse dataset which involves more sensitive attributes (western / eastern). Note that our method mainly relies on the reward signal from the validation set, which is smaller and easier to collect.
>
> **2/3 OF REPLY**

---

> ### Author Response · Authors · 2020-11-25
> **Reply to Reviewer 4, part 3/3**
>
> ### Q11. Teacher behavior
> ---
> The reason why the sampler uses so many counterfactual data is the extremely biased distribution of the CelebA dataset. In our experiments, the target output is highly correlated to the binary sensitive attribute. The distribution is defined as follows: P(A=0, Y=1) : P(A=1,Y=1) = 1 : k, where A and Y represent the protected attributes and output of interest respectively. Empirically, we found when the k is relatively large, the sampler tends to alleviate the intrinsic bias by using many counterfactual samples to reverse the distribution: P(A’=0, Y=1) : P(A’=1,Y=1) ≈ k : 1. We add experiments in section 4.2 to show the performance of two different settings:
>
> | Method                  |            Error(%)       |      EO |
> | --- | --- | --- |
> | Base5: fix ratio         |           20.7      |        0.242 |
> | Base6: reverse         |          19.2       |        0.171 |
>
> As shown in the results, The reverse setting (baseline 6), which maintains a part of the real images and adds counterfactual images to reverse the distribution (1:k to k:1) in the training data preprocessing process, could achieve a lower EO compared to other baselines. However, if we used all fake data (baseline 4), the student would never have seen the real image in the training process. Training with all synthetic data does have some limitations. Although these counterfactual images are very authentic to human eyes, there are properties that the synthetic data could not recreate and need to be learned from the real data. In conclusion, the teacher will feed slightly more real data to students to learn these patterns in the beginning and increase the amount of counterfactual data later to strengthen the reverse bias.

---

### Official Review · AnonReviewer3 · 2020-10-30
**Review of GFT**

**Rating:** 5
**Confidence:** 4

**Review:**

This paper combines counterfactual modeling with adversarial training for fair machine learning tasks. For a given fairness metric chosen from a variety of canonical examples, the method ensures fairness by augmenting the data with counterfactual examples during training. The approach has potential, which is best demonstrated on examples where the counterfactual data generation is interesting, like the CelebA data.

I believe the main weakness of the paper is a low degree of novelty. For example, the idea of training with counterfactual data is present already in the Kusner et al. (2017) reference. The current paper expands the uses of that technique and combines it with an adversarial training architecture. So the strength of this work depends on the suitability of the counterfactual model and training architecture.

The other data examples, Adult and COMPAS, do not contribute much additional value, particularly in light of my previous point. The paper could be improved by replacing these with one or more examples that better leverage the strength of adversarial training. For ICLR it might be best if these examples use types of data where representations can be useful, like text for example.

The Section 2.2 comments on in/pre/post-processing were confusing. The method is "related" to all of them? Does it not fall into any of the categories?

---

> ### Author Response · Authors · 2020-11-25
> **Reply to Reviewer 3**
>
> We would like to thank R3 for the constructive comments, and we have addressed them accordingly as follows.
>
> ### Novelty
> ---
> Our main novelty lies in the fact that we adopt a learned generator to synthesize counterfactual examples, which is optimized by a student - teacher framework. We agree with the reviewer that the idea of mitigating biases using counterfactual examples already exists in Kusner et al. (2017). However, there are several key differences between our method and Kusner et al. (2017). 1) Our counterfactual examples are generated using a powerful generator rather than a fixed synthesizer in Kusner et al. (2017). Such an improvement makes it possible to adopt our method on more sophisticated settings such as vision recognition illustrated in the CelebA dataset. The method used in Kusner et al. (2017) is difficult to work beyond tabular dataset. 2) Kusner et al. (2017) uses a fixed k : 1 ratio in training their model, where k represents the proportion between the majority class and the minority class. Our method learns the optimal ratio as training progresses and is different for each dataset. The experimental results in the revised paper (baseline 6 in Figure 4) illustrated that our method largely out-performs the baseline using a fixed ratio and on with a reversed ratio (i.e., k:1 ratio). We note that the latter is used in the Kusner et al. (2017) and is one of the drawbacks of their method compared to ours.
>
> ### Applications on datasets other than Adult and COMPAS
> ---
> We agree with the reviewer that tabular data such as Adult and COMPAS do not provide much value in demonstrating our method. As a matter of fact, we have experimented with our method using the CelebA dataset in order to demonstrate the strength of our method beyond the tabular dataset. We also agree with the reviewer that text data could be another venue to demonstrate our model. However, we believe our existing results on the CelebA dataset can also be a good indicator of the applicability of our method.
>
> ### Categorization of the method
> ---
> We believe that our method is a cross category one and relates to all of the three major fairness methods.
>
> We are considered a pre-processing method for fairness because we actively manipulate the distribution of the training data. Methods fall into this category includes the resampling or optimized pre-processing (Calmon 2017) that aims at re-adjusting the distribution of a biased dataset.
>
> We are considered an In-processing method for fairness as we directly optimize the model to achieve fairness. Methods fall into this category includes equalized odds and demographic parity. As our method builds on a fairness goal, which we rely on as a reward function in our Generative Fairness Teaching framework, we believe our method is related to in-processing.
>
> And finally, we are related to the Post-processing method because our teacher model can take any biased student model and train it to be fair. Methods falling into this category includes the learning to defer (Madras 2018) method which mitigates a biased model.

---

### Official Review · AnonReviewer1 · 2020-10-31
**Interesting and novel proposal, surprised part of it works and seeking more details**

**Rating:** 6
**Confidence:** 3

**Review:**

The paper proposes to pair a GAN based model for generating counterfactual samples given protected attribute labels and a reinforced data sampler for choosing whether to let a model train on generated data or original data.   The generative model in of itself is interesting, combining a gan component, and VAE component for analyzing an input image and a mutual information penalty for the VAE hidden vector and protected attribute.  The key to the method seems to be a reinforced data sampler, which picks, when to use a counterfactual sample versus the original sample. Given that the core of what is making this work is the data sampler, I wish there were more details.

How is the policy parameterized beyond the inputs?
How many episodes run out to tune the sampler?
The paper says the reward was the fairness measure on the held out set. Do you mean a different held out set or the same final evaluation set?
I am confused by Figure 6. Is the y-axis the % of samples that left unmodified? If so, then seems like the sampler is essentially saying always use a counterfactual? So I would expect results very similar to all-fake baseline, but in terms of EO, the results are very different. If only a sample number of samples are different than that baseline, how is this possible?

Positives:
+ the paper proposes an interesting way to generate counterfactual samples
+ the results seem promising

Negatives:
- mostly I am left confused how the reinforced data selector is making the method work. Answering some of the above questions would help.

---

> ### Author Response · Authors · 2020-11-25
> **Reply to Reviewer 1**
>
> Thank you for your constructive comments! Please see our detailed response below.
>
> ### How is the policy parameterized beyond the inputs?
> ---
> Please refer to section 3.4 for the parameterization details. To clarify the parametrization of the policy network, we have added a new paragraph in Appendix of the revised paper. Please refer to the subsection ‘‘Teacher and student model settings’ for a detailed demonstration of the parametrization of the policy network. The inputs include labels, sensitive attributes, cross entropy and the group fairness on current training batch. The policy is then predicted based on the inputs with the REINFORCE algorithm.
>
> ### How many episodes run out to tune the sampler?
> ---
> We train the teacher sampler for 500 episodes, within each episode, the student model is re-initialized and trained for 20 epochs.
>
> ### Do you mean a different held out set or the same final evaluation set?
> ---
> Officially, CelebA dataset is partitioned into training, validation, testing sets. Images 1-162770 are training, 162771-182637 are validation, 182638-202599 are testing. The reward during the training process was measured on the held-out validation set, the final result reported in table 3 was measured on the testing set.
>
> ### confused by Figure 6
> ---
> The y-axis is indeed the % of samples that left unmodified (the original samples). The reason why the sampler uses so many counterfactual data is the extremely biased distribution of the CelebA dataset. In our experiments, the target output is highly correlated to the binary sensitive attribute. The distribution is defined as follows: P(A=0, Y=1) : P(A=1,Y=1) = 1 : k, where A and Y represent the protected attributes and output of interest respectively. Empirically, we found when the k is relatively large, the sampler tends to alleviate the intrinsic bias by using many counterfactual samples to reverse the distribution: P(A’=0, Y=1) : P(A’=1,Y=1) ≈ k : 1. In order to further verify that our proposed method can outperform those simple heuristics by reversing the ratio, we’ve added additional experiments in section 4.2 to show the performance of two different baseline settings:
>
>
> | Method     |       Error(%)       |     EO |
> | --- | --- | --- |
> | Base5: fix ratio     |   20.7        |    0.242 |
> | Base6: reverse             |      19.2          |     0.171 |
>
> The first baseline we want to show is the one with reverse weight. As shown in the results, The reverse setting (baseline 6), which maintains a part of the real images and adds counterfactual images to reverse the distribution (1:k to k:1) in the training data preprocessing process, could achieve a lower EO compared to other baselines. However, if we used all fake data (baseline 4), the student would never have seen the real image in the training process. Training with all synthetic data does have some limitations. Although these counterfactual images are very authentic to human eyes, there are properties that the synthetic data could not recreate and need to be learned from the real data. The second baseline we want to show is the In the fix ratio setting(baseline 5) with very little real image. We fix the original-to-counterfactual ratio at each batch, which has 10% original image and 90% counterfactual image, the EO will also decrease by a large margin compared to the all fake setting (baseline 4).

---

### Decision · Program_Chairs · 2021-01-07
**Final Decision**

**Decision:**

Reject

**Comment:**

The paper addresses counterfactual fairness learning using generative approach. While acknowledging the importance and potential usefulness of generative approach, the reviewers and AC raised several important concerns that place this paper below the acceptance bar:

(1) low degree of novelty – see multiple concerns and suggestions by R2, R3, R4;

(2) the model is not justified by a causal mechanism (R4), and it remains unclear under which condition the proposed GAN approach is ensured to obtain unbiased counterfactual samples (R2);

 (3) lack of technical rigor when presenting the model – see R4’s request to relate to the DAG models, see R1 multiple questions regarding the reinforced data sampler;

(4) lack of empirical evidence (R3) and evaluation details, e.g. on cross validation and more recent methods (see R4’s recommendations);

 (5) related work is not discussed in sufficient details – see R4’s elaborate comment.

Among these, (4,5) did not have a substantial impact on the decision but would be helpful to address in a subsequent revision. However, (1), (2) and (3) make it very difficult to assess the benefits of the proposed approach and were viewed by AC as critical issues.
In the rebuttal it is stated that ‘Our counterfactual examples are generated using a powerful generator rather than a fixed synthesizer in Kusner et al. (2017)’ – more rigorous comparison has to be provided to support such statement. AC would urge the authors to contrast and compare their synthetic counterfactual examples with Kusner et al on the datasets where causal graph has been built. [Razieh Nabi and Ilya Shpitser. Fair inference on outcomes., AAAI2018], Figure 2 postulates causal graphs for the Compas and Adult Income datasets evaluated in this paper.

A general consensus among reviewers and AC suggests, in its current state the manuscript is not ready for a publication. We hope the detailed reviews and encouragements are useful for revising the paper.